# Sequential neuromodulation of Hebbian plasticity offers mechanism for effective reward-based navigation

**Zuzanna Brzosko[1†], Sara Zannone[2†], Wolfram Schultz[1], Claudia Clopath[2*‡], Ole Paulsen[1*‡]**

[1]Department of Physiology, Development and Neuroscience, Physiological Laboratory, Cambridge, United Kingdom; [2]Department of Bioengineering, Imperial College London, South Kensington Campus, London, United Kingdom

**Abstract** Spike timing-dependent plasticity (STDP) is under neuromodulatory control, which is correlated with distinct behavioral states. Previously, we reported that dopamine, a reward signal, broadens the time window for synaptic potentiation and modulates the outcome of hippocampal STDP even when applied after the plasticity induction protocol (Brzosko et al., 2015). Here, we demonstrate that sequential neuromodulation of STDP by acetylcholine and dopamine offers an efficacious model of reward-based navigation. Specifically, our experimental data in mouse hippocampal slices show that acetylcholine biases STDP toward synaptic depression, whilst subsequent application of dopamine converts this depression into potentiation. Incorporating this bidirectional neuromodulation-enabled correlational synaptic learning rule into a computational model yields effective navigation toward changing reward locations, as in natural foraging behavior. Thus, temporally sequenced neuromodulation of STDP enables associations to be made between actions and outcomes and also provides a possible mechanism for aligning the time scales of cellular and behavioral learning.

*For correspondence: c.clopath@imperial.ac.uk (CC); op210@cam.ac.uk (OP)

[†]These authors contributed equally to this work
[‡]These authors also contributed equally to this work

## Introduction

Spike timing-dependent plasticity (STDP) is a form of Hebbian learning that depends on the order and precise timing of presynaptic and postsynaptic spikes (*Gerstner et al., 1996*; *Markram et al., 1997*; *Bi and Poo, 1998*; *Song et al., 2000*). STDP is a computationally attractive mechanism that has been implicated in several forms of learning and memory including competitive Hebbian learning (*Song et al., 2000*; *Clopath et al., 2010*). Nevertheless, from a theoretical perspective, two properties complicate its use as a general mechanistic explanation of behavioral learning. Firstly, in contrast to many forms of behavioral learning, conventional STDP is unsupervised, that is, the resulting plasticity is not informed by the outcome of the activity (*Hertz et al., 1991*; *Gerstner et al., 2014*). Secondly, STDP relies on shorter time scales than most behaviors. Given that STDP is under neuromodulatory influence (*Seol et al., 2007*; *Zhang et al., 2009*; *Pawlak et al., 2010*), here we sought to determine whether behaviorally relevant activation of different neuromodulatory signals could address both of these issues in hippocampus-dependent learning.

A wealth of experimental data has implicated the hippocampus with its constituent place cells (*O'Keefe and Dostrovsky, 1971*; *O'Keefe and Nadel, 1978*) in spatial learning and memory in non-human animals (*Morris et al., 1982*) as well as humans (*Ekstrom et al., 2003*). Hippocampal synaptic plasticity (*Bliss and Lomo, 1973*; *Bliss and Collingridge, 1993*), including STDP (*Bi and Poo, 1998*; *Debanne et al., 1998*; *Kwag and Paulsen, 2009*; *Andrade-Talavera et al., 2016*), is believed to mediate the encoding of spatial memories (*Morris et al., 1986*; *Tsien et al., 1996*).

Different neuromodulatory inputs into the hippocampus are active during distinct behavioral states. In particular, whilst exploratory behavior coincides with an increased cholinergic tone (*Kametani and Kawamura, 1990*; *Marrosu et al., 1995*; *Thiel et al., 1998*), rewards are associated with activity of dopaminergic neurons (*Schultz et al., 1997*; *Suri and Schultz, 1999*; *Pan et al., 2005*). Importantly, we recently demonstrated that dopamine can convert hippocampal timing-dependent long-term depression (t-LTD) into timing-dependent long-term potentiation (t-LTP) when applied after a delay of minutes (*Brzosko et al., 2015*), thus providing a possible mechanism for a supervisory signal to associate specific experiences with delayed rewards in hippocampus-dependent reward learning. Here, we set out to investigate the effect of cholinergic and subsequent dopaminergic activity on hippocampal STDP. Moreover, we wanted to develop and test an ethologically relevant computational model of reward-based spatial navigation based on this temporally sequenced neuromodulation of STDP. Specifically, we wished to establish whether the temporal characteristics of cholinergic and dopaminergic modulation of hippocampal synaptic plasticity can explain key aspects of adaptive foraging behavior in a changing environment such as exploration and reward-seeking navigation, including unlearning of previously exploited reward locations.

## Results and discussion

To examine the physiological rules governing hippocampal STDP, we first wanted to investigate the effect of cholinergic modulation on the induction of synaptic plasticity. As before (*Brzosko et al., 2015*), we monitored excitatory postsynaptic potentials (EPSPs) evoked by extracellular stimulation of the Schaffer collateral pathway during whole-cell recordings of CA1 pyramidal cells in mouse horizontal slices (postnatal days 12–18; Materials and methods). Plasticity was induced in current clamp mode using an induction protocol consisting of 100 pairings of a single EPSP followed by a single postsynaptic spike (pre-before-post pairing) or a single postsynaptic spike followed by a single EPSP (post-before-pre pairing) at 0.2 Hz. Consistent with previous studies (*Bi and Poo, 1998*; *Zhang et al., 2009*; *Edelmann and Lessmann, 2011*; *Brzosko et al., 2015*), in control condition the pre-before-post pairing with a time interval between the presynaptic and postsynaptic activity ($\Delta t$) of 10 ms induced t-LTP (+ Pairing – ACh: $135 \pm 7\%$, $t(7) = 4.7$, p=0.0022 vs. 100%, $n = 8$; *Figure 1a,b*; *Figure 1—source data 1*). However, when acetylcholine (1 µM) was bath-applied for 10 min from 1 to 2 min before and during the same pre-before-post pairing in interleaved experiments, robust t-LTD was induced (+ Pairing + ACh: $63 \pm 8\%$, $t(5) = 4.9$, p=0.0046 vs. 100%, $n = 6$; $t(12) = 6.6$, p<0.0001 vs. + Pairing – ACh; *Figure 1a,b*; *Figure 1—source data 1*). To exclude the possibility that acetylcholine by itself could induce LTD in the test pathway, control experiments with ongoing synaptic stimulation over 60 min at 0.2 Hz, but without pairing with postsynaptic action potentials, were performed. In accordance with earlier reports (*Sugisaki et al., 2011*), application of acetylcholine for 10 min had no significant effect on the basal Schaffer collateral transmission (– Pairing + ACh: $90 \pm 6\%$, $t(6) = 1.8$, p=0.1220 vs. 100%, $n = 7$; *Figure 1a,b*; *Figure 1—source data 1*). Since muscarinic acetylcholine receptors are highly expressed in the hippocampus (*Levey et al., 1995*), the specificity of this cholinergic modulation of STDP was assessed using the muscarinic acetylcholine receptor antagonist atropine (100 nM). The application of atropine prevented acetylcholine-facilitated t-LTD, resulting in significant t-LTP instead ($\Delta t$ =+10 ms; +ACh: $63 \pm 9\%$, $t(4) = 4.1$, p=0.0146 vs. 100%, $n = 5$; +ACh + Atropine: $141 \pm 8\%$, $t(5) = 5.1$, p=0.0037 vs. 100%, $n = 6$; $t(9) = 6.5$, p<0.0001 vs. +ACh; *Figure 1c,d*; *Figure 1—source data 1*). This implies that the facilitation of t-LTD was due to specific muscarinic acetylcholine receptor activation. Input-specific t-LTD was also induced when acetylcholine was applied during the pairing protocol with $\Delta t = 0$ ms (Control: $160 \pm 17\%$, $t(4) = 3.6$, p=0.0238 vs. 100%, $n = 5$; + ACh: $64 \pm 13\%$, $t(7) = 2.9$, p=0.0245 vs. 100%, $n = 8$; $t(4) = 5.9$, p=0.0041 vs. Control; *Figure 1e*; *Figure 1—source data 1*; unpaired control pathway not shown) and $\Delta t = -20$ ms (Control: $66 \pm 10\%$, $t(6) = 3.5$, p=0.0124 vs. 100%, $n = 7$; + ACh: $63 \pm 12\%$, $t(5) = 3.0$, p=0.0292 vs. 100%, $n = 6$; $t(5) = 0.7$, p=0.5265 vs. Control; *Figure 1e*; *Figure 1—source data 1*). However, pairing protocols with spike timing interval extended to ±50 ms during application of acetylcholine did not lead to a significant change in synaptic weights ($\Delta t$ =+50 ms: $84 \pm 11\%$, $t(6) = 1.5$, p=0.1852 vs. 100%, $n = 7$; $\Delta t = -50$ ms: $113 \pm 15\%$, $t(5) = 0.8$, p=0.4409 vs. 100%, $n = 6$; *Figure 1e*; *Figure 1—source data 1*). Together, our results suggest that the activation of muscarinic acetylcholine receptors during the coordinated spiking activity biases STDP toward synaptic depression at the Schaffer collateral pathway. Hence, both dopamine (*Brzosko et al., 2015*) and

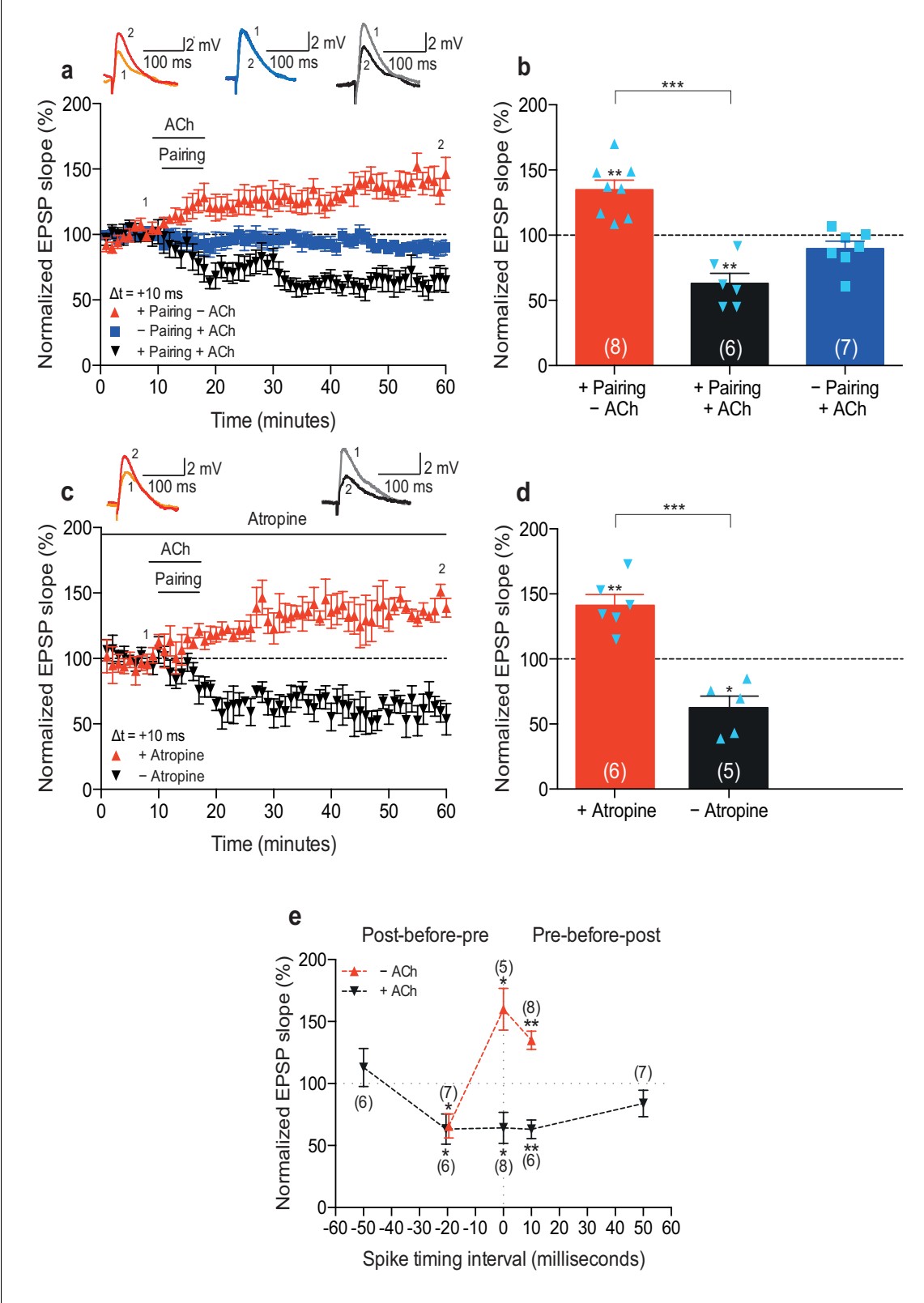

**Figure 1.** Acetylcholine biases STDP toward depression. (**a**) In control condition, the pre-before-post pairing protocol with *Δ*t =+10 ms induced t-LTP (red), whereas in the presence of 1 μM acetylcholine (ACh), the same pairing protocol induced t-LTD (black). In the absence of the pairing protocol, ACh had no effect on baseline EPSPs (blue). Traces show an EPSP before (1) and 40 min after pairing (2) for each condition. (**b**) Summary of results (mean ± s.e.m.). (**c**) Application of muscarinic ACh receptor antagonist, 100 nM atropine, at the beginning of the recordings prevented ACh-facilitated

*Figure 1 continued on next page*

*Figure 1 continued*

t-LTD (Δt =+10 ms; black) and pre-before-post pairing resulted in significant t-LTP (red). Traces are presented as in **a**. (**d**) Summary of results. (**e**) Summary of the STDP induction with various spike timing intervals (Δt in ms) in control condition (− ACh; red) and in the presence of ACh (+ ACh; black). Each data point is the group mean percentage change from baseline of the EPSP slope. Error bars represent s.e.m. Significant difference (*p<0.05, **p<0.01, ***p<0.001) compared with the baseline or between the indicated two groups (two-tailed Student's *t*-test). The total numbers of individual cells (blue data points) are shown in parentheses.

The following source data and figure supplements are available for figure 1:

**Source data 1.** Source data for *Figure 1*.
**Source data 2.** Source data for *Figure 1—figure supplement 1*.
**Source data 3.** Source data for *Figure 1—figure supplement 2*.
**Figure supplement 1.** Neuromodulation of STDP by dopamine and co-application of dopamine and acetylcholine.
**Figure supplement 2.** Low concentration of acetylcholine prevents development of t-LTP.

---

acetylcholine (*Figure 1*) modulate synaptic plasticity to yield synaptic potentiation (*Figure 1—figure supplement 1a,c*; *Figure 1—source data 2*; *Brzosko et al., 2015*) and depression (*Figure 1e*), respectively, irrespective of the precise spike order during pairing. Each of the two neuromodulators effectively converts a spike timing-dependent learning rule into a correlation-based learning rule but they do so in opposite directions. We also investigated the effect of co-application of acetylcholine and dopamine. We found that co-activation of dopaminergic and cholinergic receptors results in synaptic depression with post-before-pre pairing and leads to an initial synaptic depression followed by a gradual reversal of the synaptic weights back toward baseline with pre-before-post pairing (*Figure 1—figure supplement 1b,c*; *Figure 1—source data 2*).

While several previous studies using plasticity induction protocols other than STDP showed that acetylcholine, primarily via activation of muscarinic M1 receptors, facilitates hippocampal LTP (*Boddeke et al., 1992*; *Huerta and Lisman, 1995*; *Ovsepian et al., 2004*; *Shinoe et al., 2005*; *Buchanan et al., 2010*; *Connor et al., 2012*; *Digby et al., 2012*; *Dennis et al., 2016*), acetylcholine has also been found to induce LTD, particularly at higher concentrations (*Scheiderer et al., 2006*; *Volk et al., 2007*; *Dickinson et al., 2009*; *Jo et al., 2010*; *Kamsler et al., 2010*) — likely through the activation of a number of muscarinic receptor subtypes (*Teles-Grilo Ruivo and Mellor, 2013*). Thus, the polarity of acetylcholine-modulated plasticity can depend on the concentration of agonist used and specific cholinergic receptor subtype activated (*Müller et al., 1988*; *Auerbach and Segal, 1996*; *Dennis et al., 2016*). Under our experimental conditions, a lower concentration of acetylcholine (100 nM) during pre-before-post pairing did not result in synaptic depression, but it prevented significant potentiation from developing (Δt =+10 ms; − ACh: 162 ± 16%, *t*(5) = 3.8, p=0.0127 vs. 100%, *n* = 6; + ACh: 113 ± 9%, *t*(9) = 1.4, p=0.1888 vs. 100%, *n* = 10; *t*(14) = 2.9, p=0.0119 vs. − ACh; *Figure 1—figure supplement 2*; *Figure 1—source data 3*). Our findings are consistent with previous reports using STDP induction protocols, showing that cholinergic receptor activation enables t-LTD in the visual cortex (*Seol et al., 2007*) and facilitates t-LTD in the dorsal cochlear nucleus (*Zhao and Tzounopoulos, 2011*), but appear to contrast with results from studies in the rat hippocampus (*Adams et al., 2004*; *Sugisaki et al., 2011*, *2016*).

A crucial aspect of foraging involves reward-seeking behavior. Most reinforcement learning models rely on the ability of the reinforcing signal to strengthen active synapses, also when it arrives after the activity has occurred (*Sutton and Barto, 1981*). Therefore, we next wanted to extend our previous finding that the reinforcing signal dopamine can retroactively convert t-LTD into t-LTP (*Brzosko et al., 2015*) by investigating whether dopamine also modulates acetylcholine-facilitated t-LTD. Bath-application of atropine (100 nM) to block muscarinic acetylcholine receptors immediately after acetylcholine washout did not affect the expression of acetylcholine-facilitated t-LTD (Δt =+10 ms; 79 ± 9%, *t*(9) = 2.4, p=0.0399 vs. 100%, *n* = 10; *Figure 2a,c*; *Figure 2—source data 1*). Atropine also did not significantly affect baseline EPSPs (83 ± 8%, *t*(6) = 2.2, p=0.0747 vs. 100%, *n* = 7;

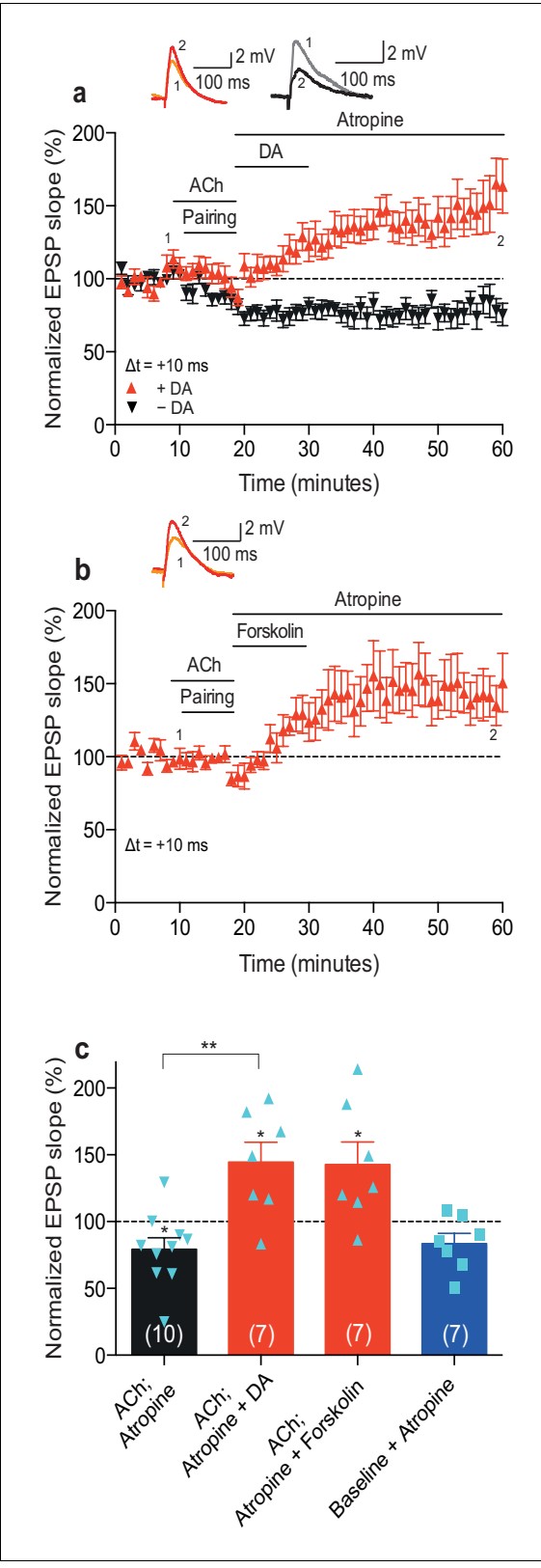

**Figure 2.** Dopamine retroactively converts acetylcholine-facilitated t-LTD into t-LTP. (a) Application of muscarinic acetylcholine receptor antagonist, 100 nM atropine, following acetylcholine (ACh; 1 µM) washout after the pre-before-post pairing protocol with Δt = +10 ms did not affect the development of ACh-facilitated t-LTD (black). Dopamine (DA; 100 µM) applied, together with atropine, immediately after ACh washout at the end of the same

*Figure 2 continued on next page*

*Figure 2 continued*

pairing protocol, converted ACh-facilitated t-LTD into t-LTP (red). Traces show an EPSP before (1) and 40 min after pairing (2) in the two conditions. (**b**) Forskolin (50 μM), applied together with atropine, converted ACh-facilitated t-LTD into t-LTP, mimicking the effect of DA. Traces are presented as in a. (**c**) Summary of results from a and b. In the absence of the pairing protocol, atropine had no significant effect on baseline EPSPs (blue). Error bars represent s.e.m. Significant difference (*p<0.05) compared with the baseline or between the indicated two groups (two-tailed Student's *t*-test). The total numbers of individual cells (blue data points) are shown in parentheses.

The following source data and figure supplement are available for figure 2:

**Source data 1.** Source data for *Figure 2*.
**Source data 2.** Source data for *Figure 2—figure supplement 1*.
**Figure supplement 1.** Acetylcholine applied after the pairing protocol does not affect t-LTP.

---

*Figure 2c*; *Figure 2—source data 1*). Notably, when, in the presence of atropine (100 nM) to prevent sustained muscarinic receptor activation, dopamine (100 μM) was added to the superfusion fluid, for 10–12 min starting within 1 min after the pairing protocol in interleaved experiments, it converted acetylcholine-facilitated t-LTD into t-LTP (Δt =+10 ms; ACh + Atropine + DA: 145 ± 15%, *t*(6) = 3.0, p=0.0244 vs. 100%, *n* = 7; *t*(15) = 4.0, p=0.0011 vs. ACh + Atropine; *Figure 2a,c*; *Figure 2—source data 1*). Similarly, the adenylyl cyclase activator forskolin (50 μM), applied for 10–12 min immediately after the pairing protocol with Δt =+10 ms, also resulted in robust conversion of acetylcholine-facilitated t-LTD into t-LTP, emulating the effect of dopamine (143 ± 17%, *t*(6) = 2.5, p=0.0447 vs. 100%, *n* = 7; *Figure 2b,c*; *Figure 2—source data 1*). The magnitudes of the resultant synaptic potentiation in the current study (*Figure 2a,b*; *Figure 2—source data 1*) are comparable to those of the dopamine-induced and forskolin-induced conversion of t-LTD into t-LTP in our previous study (Δt = −20 ms; + DA: 154 ± 10%, *t*(10) = 5.2, p=0.0004 vs. 100%, *n* = 11; + Forskolin: 167 ± 17%, *t*(6) = 3.9, p=0.0078 vs. 100%, *n* = 7; Figure 4a,b in *Brzosko et al., 2015*). These results show that dopamine can convert not only post-before-pre pairing-induced t-LTD (*Brzosko et al., 2015*) but also acetylcholine-facilitated pre-before-post pairing-induced t-LTD into t-LTP, when acting within a few minutes following the induction protocol. The effect of dopamine is therefore irrespective of the precise spike order and is mediated at least in part via the activation of the cyclic adenosine monophosphate (cAMP) signaling cascade. In contrast, acetylcholine did not have an effect on plasticity when applied after the induction protocol (*Figure 2—figure supplement 1*; *Figure 2—source data 2*).

At the synaptic level, our data emphasize three crucial features of neuromodulation of hippocampal STDP: (i) neuromodulation enables correlation-based synaptic plasticity, with dopamine promoting potentiation and acetylcholine facilitating depression (*Figure 1*); (ii) synaptic weights are not only graded in magnitude but also can shift polarity by subsequent neuromodulation (*Figures 1* and *2*; *Brzosko et al., 2015*); and (iii) the conversion of synaptic depression into potentiation by dopamine can occur even after extended delay (*Figure 2*; *Brzosko et al., 2015*). These characteristics suggest that sequential neuromodulation of hippocampal STDP is behaviorally relevant (*Figure 3a*). We therefore wanted to test possible functional implications of our newly uncovered synaptic learning rule in a computational model of hippocampus-dependent reward-based navigation (*Figure 3bi*). We developed a network model with place cells that code for the location of an agent in its environment. These place cells projected onto action neurons that were part of a winner-takes-all network (*Figure 3bii*) and dictated the speed and direction of the agent (*Figure 3biii, biv*; *Foster et al., 2000*; *Vasilaki et al., 2009*; *Frémaux et al., 2010*, *2013*; Materials and methods). The synaptic connections between the place cells and the action neurons were subject to STDP. In test simulations (*Figure 4a*: + ACh), they followed our novel sequentially neuromodulated synaptic learning rule, with synaptic weights being updated in all trials. The weights were potentiated when reward, signaled via dopamine (*Schultz et al., 1997*; *Suri and Schultz, 1999*; *Pan et al., 2005*), was found before the end of a trial but they were depressed when reward was not found, reflecting the effect of the increased cholinergic tone associated with exploration (*Kametani and Kawamura,*

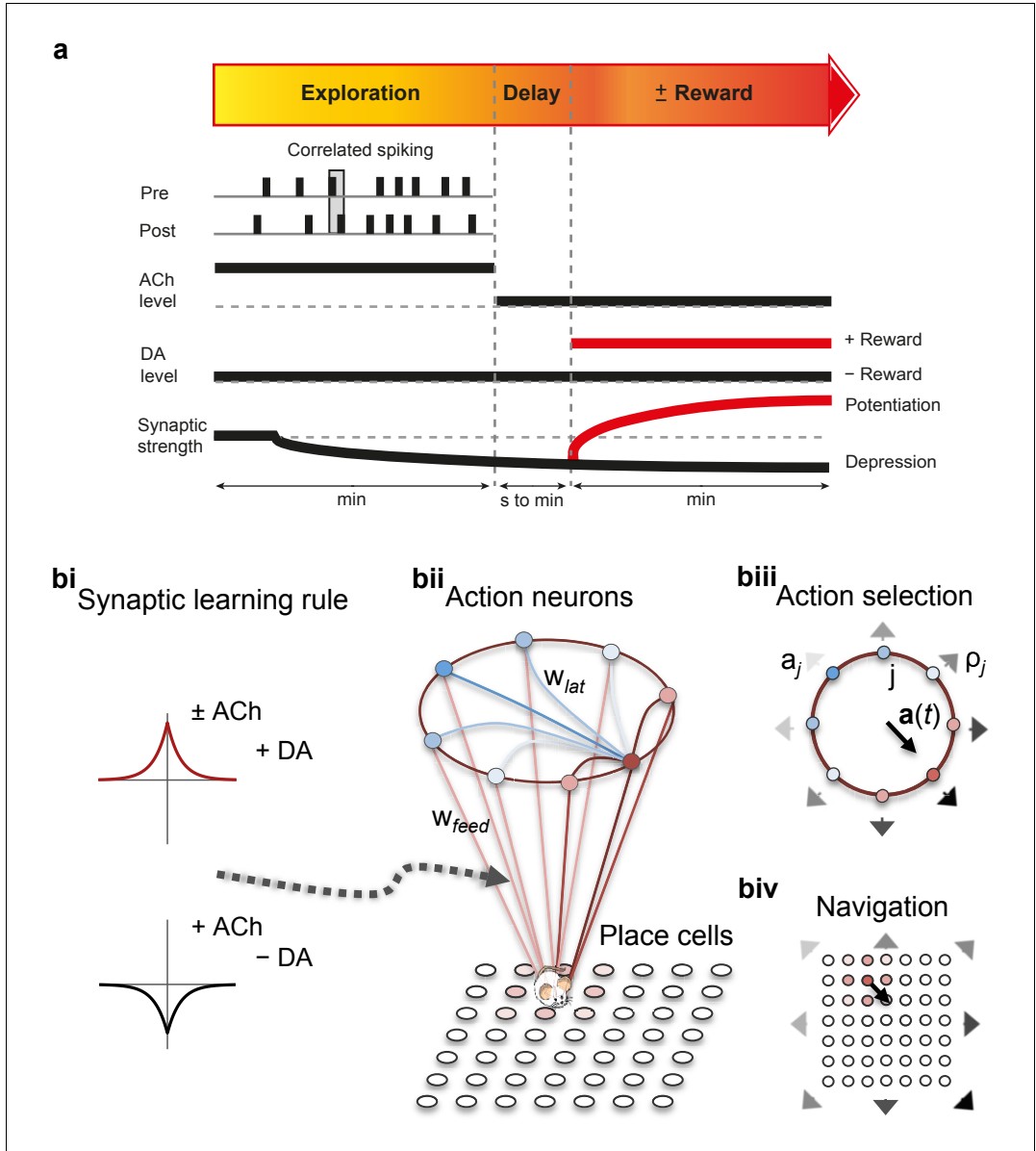

**Figure 3.** From plasticity to behavior: A computational model. (**a**) Schematic diagram of synaptic and behavioral timescales in reward-related learning. During Exploration, the activity-dependent modification of synaptic strength due to spike timing-dependent plasticity (STDP) depends on the coordinated spiking between presynaptic and postsynaptic neurons on a millisecond time scale. STDP develops gradually on a scale of minutes. Increased cholinergic tone (ACh) during Exploration facilitates synaptic depression. When Reward, signaled via dopamine (DA), follows Exploration with a Delay of seconds to minutes, synaptic depression is converted into potentiation. (**b**) Computational model. (**bi**) Symmetric STDP learning windows incorporated in the model, where acetylcholine biases STDP toward synaptic depression, while subsequent application of dopamine converts this depression into potentiation. (**bii**) The position of the agent in the field, **x**(*t*), is coded by place cells and its moves are determined by the activity of action neurons. STDP is implemented in the feed-forward connections between place cells and action neurons. Place cells become active with the proximity of the agent (active neurons in red: the darker, the higher their firing rate). Place cells are connected to action neurons through excitatory synapses (w$_{feed}$: the darker, the stronger the connection). Action neurons are connected with each other: recurrent synaptic weights (w$_{lat}$) are excitatory (red) when action neurons have similar tuning, or inhibitory (blue) otherwise. Thus, the activation of action neurons is dependent on both the feed-forward and recurrent connections. (**biii**) Each action neuron *j* codes for a different direction a$_j$ (large arrow's direction) and has a different firing rate ρ$_j$ (large arrow's color: the darker, the higher the firing rate). The action to take a(*t*) (black arrow) is the average of all directions, weighted by their respective firing rate. (**biv**) The agent takes action a(*t*). Therefore, it moves to x(*t* + *Δt*) = x(*t*) + a(*t*).

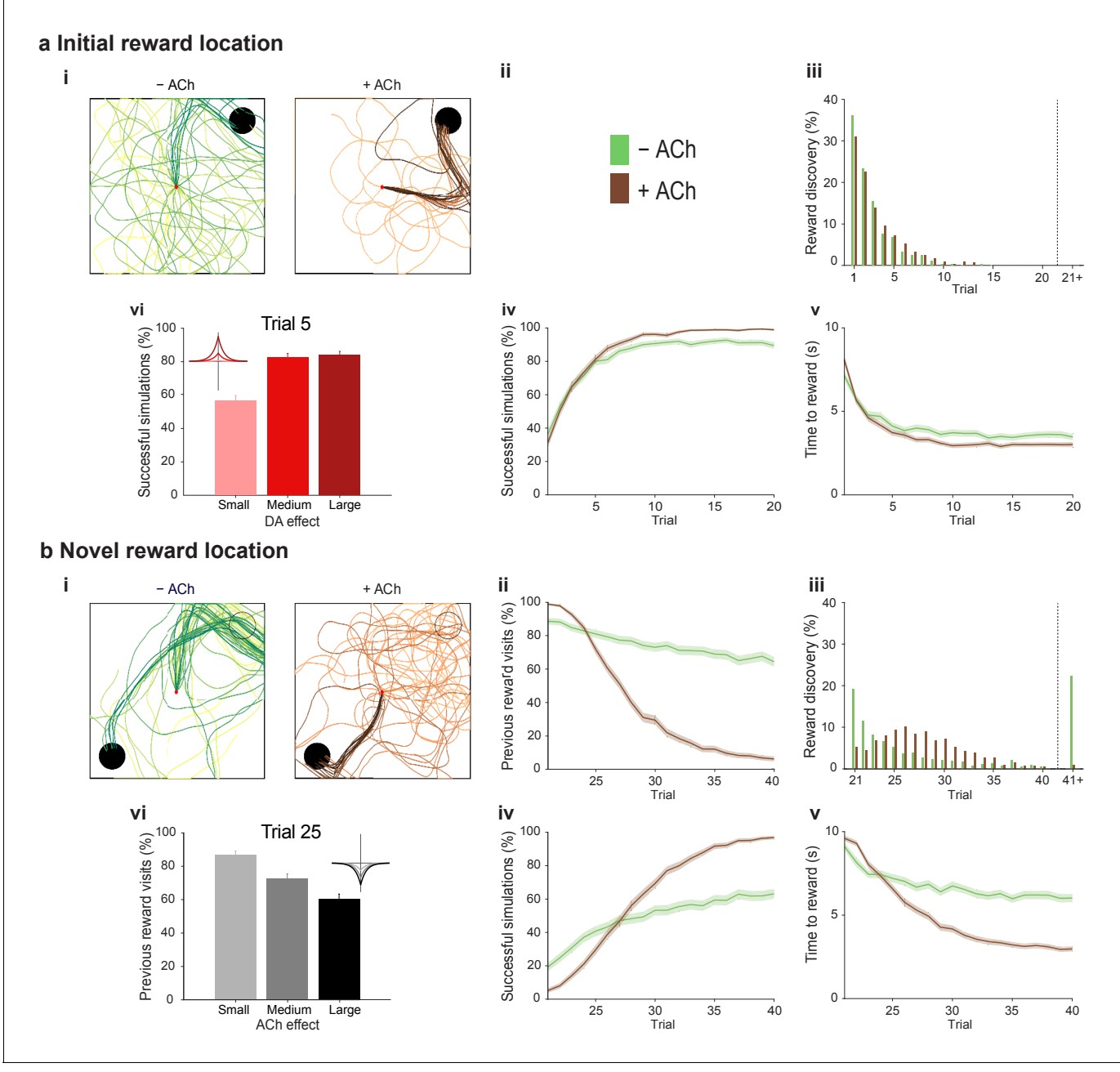

**Figure 4.** Temporally sequenced cholinergic and dopaminergic modulation of STDP yields effective navigation toward changing reward locations. (**a**) Learning of an initial reward location (trials 1–20; 1000 simulations in each trial) shows a modest improvement in learning when cholinergic depression is included in the model. (**i**) Example trajectories. The agent starts from the center of the open field (red dot) and learns the reward location (closed black circle) with (+ ACh; brown) and without (− ACh; green) cholinergic depression built into the model. Trials are coded from light to dark, according to their temporal order (early = light, late = dark). (**ii**) Color scheme. (**iii**) Reward discovery. The graph shows percent cumulative distribution of trials in which the reward location is discovered for the first time. (**iv**) Learning curve presented as a percentage of successful simulations over successive trials. (**v**) Average time to reward in each successful trial. Unsuccessful trials, in which the agent failed to find the reward, were excluded. (**vi**) Percentage of successful simulations in trial 5, under conditions with different magnitudes of dopaminergic effect (learning windows in the top-left corner). Decreasing the magnitude of dopaminergic potentiation significantly affects learning (p<0.001, two-sample Student's *t*-test: Small vs. Medium and Small vs. Large). Under Medium and Large conditions, the agent performs similarly most likely due to a saturation effect (p>0.05, two-sample Student's *t*-test: Medium vs. Large). (**b**) Learning of a displaced reward location is facilitated when cholinergic depression is included in the model. (**i**) Example trajectories (trials 21–40; 1000 simulations in each trial). The agent learns a novel reward location (closed circle; previously exploited reward = open circle). Trajectories presented as in **ai**. Comparison of control (− ACh) and test (+ ACh) simulations: (**ii**) visits to previous reward location (%); (**iii**) trial number at novel reward discovery; (**iv**) successful reward collection over successive trials (%); (**v**) average time to reward over trials. (**vi**) Percentage of visits to the old reward

*Figure 4 continued*

location in trial 25, under conditions with different magnitudes of cholinergic depression (learning windows in the top-right corner). Increasing the magnitude of acetylcholine effect yields faster unlearning (p<0.001, two-sample Student's *t*-test: Small vs. Medium, Medium vs. Large and Small vs. Large). The graphs (**biii-bv**) are presented as in **a**. The shaded area (**aiv-v** and **bii**, **biv-v**) represents the 95% confidence interval of the sample mean.

The following figure supplements are available for figure 4:

**Figure supplement 1.** Exploration following reward displacement.

**Figure supplement 2.** The magnitude of dopamine effect affects learning.

**Figure supplement 3.** The magnitude of acetylcholine effect affects unlearning.

**Figure supplement 4.** The integral of the asymmetric STDP learning window determines the performance of the agent.

*1990*; *Marrosu et al., 1995*; *Thiel et al., 1998*). These test simulations were compared against a second set of simulations (*Figure 4a*: − ACh), with reward-modulated STDP following a positive symmetric window (*Figure 3bi*), gated by dopamine and with no interaction with acetylcholine, as in previous computational work (*Florian, 2007*; *Izhikevich, 2007*; *Legenstein et al., 2008*). In this control set of simulations, the weights were potentiated when the reward was found before the end of a trial, but they were unchanged otherwise. Consistent with previous reward-modulated STDP models (*Frémaux et al., 2010*, *2013*), dopaminergic potentiation following a successful trial, in which the agent found the reward, enabled the agent to progressively develop effective navigation toward the reward location (trials 1–20; *Figure 4ai,aii*). During this initial reward learning phase, implementing the sequential modulation by acetylcholine and dopamine in the model led to a modest improvement in learning compared to control simulations (reward discovery: *Figure 4aiii*; successful trials: *Figure 4aiv*; time to reward: *Figure 4av*). The functional advantage of cholinergic depression became evident in the second phase of the simulation, when the reward location was moved to the opposite corner of the open field (*Figure 4bi*). In this case, cholinergic depression enabled unlearning of the previously rewarded location (*Figure 4bii*), which led to enhanced exploration (*Figure 4—figure supplement 1*), thereby facilitating subsequent learning of the new reward location (*Figure 4biii–v*). This is demonstrated by the large difference in the number of visits to the old reward location between the two sets of simulations (trials 21–40; + ACh vs. − ACh; *Figure 4bii*). The agent discovered the new reward location in fewer trials (*Figure 4biii*), exhibited a higher percentage of successful simulations over successive trials (*Figure 4biv*), and showed progressive reduction in the average time and trajectory length to reward in successful simulations (*Figure 4bi,bv*).

To formulate testable predictions of the relationship between cellular neuromodulation and behavior, we proceeded to vary the degree of neuromodulation in our model. Decreasing the magnitude of dopaminergic potentiation resulted in slower learning, while increasing it did not produce any statistically significant difference (*Figure 4avi* and *Figure 4—figure supplement 2*), probably due to a saturation effect. Altering the magnitude of cholinergic depression revealed a clear effect on unlearning performance, where increasing the magnitude of acetylcholine effect yielded faster unlearning (*Figure 4bvi* and *Figure 4—figure supplement 3*).

Our computational model bears resemblance to conventional reinforcement learning models (*Sutton and Barto, 1998*), in which the active connections carry a trace making them *eligible* for learning changes should a reinforcing event ('reward') occur later in time. This concept of '*eligibility trace*' is at the core of network models of reinforcement learning: it is both necessary and sufficient to learn causal associations between synaptic activity and delayed outcomes (*Florian, 2007*; *Izhikevich, 2007*; *Legenstein et al., 2008*). Not only do our experimental results offer evidence for these mechanisms, which previously had been only theorized, but they also reveal novel characteristics with important computational implications. Firstly, neuromodulation converts classical STDP into a correlation-based rule, with same-sign weight changes for both pre-before-post and post-before-pre pairings (*Brzosko et al., 2015*; this paper). This feature is incorporated into our model where, rather than the exact spike timing, it is the integral of the learning window (*Figure 4—figure supplement 4*) that determines whether the agent learns (positive window) or unlearns (negative window).

Secondly, cholinergic depression allows the agent to explicitly learn from unrewarded trials (this paper). This leads to a significant computational advantage. Our sequentially neuromodulated synaptic learning rule (test simulations), which allows the agent to learn from both rewarded (+ ACh + DA) and unrewarded actions (+ ACh – DA), outperformed both the model with symmetric dopamine-modulated learning window (– ACh + DA; *Figure 3bi*) and the standard reward modulated STDP model (*Figure 4—figure supplement 4*) when the reward location was moved. Incorporating cholinergic depression allowed the agent to appropriately shift between learning and unlearning in a task-relevant manner. Several computational functions have been previously suggested for acetylcholine (*Doya, 2002*; *Yu & Dayan, 2005*; *Hasselmo, 1999*). Our results suggest a role for acetylcholine that is complementary to that of dopamine and particularly relevant to changing environments. Indeed, as demonstrated here, a simple feed-forward network obeying cholinergic-dopaminergic sequential neuromodulation of STDP is sufficient to guide an agent to changing reward locations as required during natural foraging behavior (*Charnov, 1976*; *Stephens and Krebs, 1986*).

In conclusion, our findings demonstrate that muscarinic acetylcholine receptor activation during coordinated spiking activity biases STDP toward synaptic depression. Meanwhile, the reward signal dopamine, via activation of the cAMP pathway, converts both conventional t-LTD (*Brzosko et al., 2015*) and acetylcholine-facilitated synaptic depression into potentiation. Incorporating this synaptic learning rule into a simple feed-forward neural network model successfully guides the agent towards changing reward locations similar to natural foraging behavior. We suggest that sequential neuromodulation of synaptic plasticity provides a robust biological mechanism that might be used in reward-based navigation and other hippocampus-dependent functions.

## Materials and methods

### Electrophysiology experiments
#### Animals
The research was performed under the Animals (Scientific Procedures) Act 1986 Amendment Regulations 2012 following ethical review by the University of Cambridge Animal Welfare and Ethical Review Body (AWERB). The animal procedures were authorised under Project licence PPL 70/8892.

Wild-type mice (C57BL/6; RRID:IMSR_JAX:000664; postnatal days 12–18; from Harlan, Bicester, UK or Central Animal Facility, Cambridge University) of both sexes were housed on a 12 hr light/dark cycle at 19–23°C, with water and food *ad libitum*. Caution was taken to minimize stress and the number of animals used in experiments.

#### Slice preparation
Mice were anesthetized with isoflurane and decapitated. The brain was rapidly removed, glued to the stage of a vibrating microtome (Leica VT 1200S, Leica Biosystems, Wetzlar, Germany) and immersed in ice-cold artificial cerebrospinal fluid (ACSF) containing the following (mM): 126 NaCl, 3 KCl, 26.4 NaH$_2$CO$_3$, 1.25 NaH$_2$PO$_4$, 2 MgSO$_4$, 2 CaCl$_2$, and 10 glucose. The ACSF solution, with pH adjusted to 7.2 and osmolarity to 270–290 mOsm L$^{-1}$, was continuously bubbled with carbogen gas (95% O$_2$/5% CO$_2$). The brain was sectioned into 350-μm-thick horizontal slices. The slices were incubated in ACSF at room temperature in a submerged-style storage chamber for at least 1 hr. For recordings (1–7 hr after slicing), individual slices were transferred to an immersion-type recording chamber, perfused with ACSF (2 mL min$^{-1}$) at 24–26°C.

#### Electrophysiology
##### Whole-cell recordings
Whole-cell patch-clamp recordings were made from CA1 pyramidal neurons (located adjacent to the *stratum oriens*). For stimulation of Schaffer collaterals, a monopolar stimulation electrode was placed in the *stratum radiatum* of the CA1 subfield. The hippocampal subfields were visually identified using infrared differential interference contrast (DIC) microscopy. Patch pipettes (resistance: 4–8 MΩ) were made from borosilicate glass capillaries (0.68 mm inner diameter, 1.2 mm outer diameter), pulled using a P-97 Flaming/Brown micropipette puller (Sutter Instruments Co., Novato, California, USA). The internal solution of patch pipettes was (mM): 110 potassium gluconate, 4 NaCl, 40 HEPES, 2

ATP-Mg, 0.3 GTP (pH adjusted with 1 M KOH to 7.2, and osmolarity with ddH$_2$O to 270 mOsm L$^{-1}$). The liquid junction potential was not corrected for. Cells were accepted for experiment only if the resting membrane potential at the start of the recording was between −55 and −70 mV. Membrane potential was held at −70 mV throughout further recording by direct current application via the recording electrode. At the beginning of each recording, all cells were tested for regular spiking responses to positive current steps – characteristic of pyramidal neurons.

## Stimulation protocol

Single excitatory postsynaptic potentials (EPSPs) of amplitude between 3 and 8 mV were evoked at 0.2 Hz by adjusting the magnitude of direct current pulses (stimulus duration 50 μs, intensity 100 μA-1 mA). After a stable EPSP baseline period of at least 10 min, a pairing protocol was applied consisting of a single presynaptic EPSP evoked by stimulation of Schaffer collaterals and a single postsynaptic action potential elicited with the minimum effective somatic current pulse (1–1.8 nA, 3 ms) via the recording electrode, repeated 100 times at 0.2 Hz. Spike-timing intervals (Δt in ms) were measured between the onset of the EPSP and the onset of the action potential. The EPSPs were monitored for at least 40 min after the end of the pairing protocol. Presynaptic stimulation frequency remained constant throughout the experiment.

## Data acquisition and data analysis

Voltage signals were low-pass filtered at 2 kHz using an Axon Multiclamp 700B amplifier (Molecular Devices, Sunnyvale, California, USA). Data were acquired at 5 kHz via an ITC18 interface board (Instrutech, Port Washington, New York, USA), transmitting to a Dell computer running the Igor Pro software (RRID:SCR_000325; WaveMetrics, Lake Oswego, Oregon, USA). All experiments were carried out in the current clamp ('bridge') mode. Series resistance was monitored (10–15 MΩ) and compensated for by adjusting the bridge balance. Data were discarded if series resistance changed by more than 30%. Data were analyzed using Igor Pro. EPSP slopes were measured on the rising phase of the EPSP as a linear fit between the time points corresponding to 25–30% and 70–75% of the peak amplitude. For statistical analysis, the mean EPSP slope per minute of the recording was calculated from 12 consecutive sweeps and normalized to the baseline. Normalized ESPS slopes from the last 5 min of the baseline (immediately before pairing) and from the last 5 min of the recording (35–40 min or 55–60 min after pairing) were averaged. The magnitude of plasticity, as an indicator of synaptic change, was defined as the average EPSP slope after pairing expressed as a percentage of the average EPSP slope during baseline.

## Drugs

The following drugs were used: acetylcholine chloride 1 μM or 100 nM, atropine 100 nM, dopamine hydrochloride 20 μM, forskolin 50 μM. All drugs (purchased from Sigma-Aldrich, Dorset, United Kingdom; or Tocris Bioscience, Bristol, United Kindgom) were bath-applied through the perfusion system by dilution of concentrated stock solutions (prepared in water or DMSO) in ACSF.

## Statistical analysis

Statistical comparisons were made using one-sample or two-sample two-tailed Student's $t$-test as appropriate, with a significance level of $\alpha = 0.05$. Data are presented as mean ± s.e.m. Significance levels are indicated by *p<0.05, **p<0.01, ***p<0.001.

## Computational modeling

All computer modeling was done in Matlab (RRID:SCR_001622). The code will be posted on ModelDB after publication. The navigation model is based on a one-layer network (*Frémaux et al., 2013*). The *place cells* in the input layer code for the position of the agent in the environment. They project to the output layer of *action neurons*. Each one of the action neurons represents a different direction. The lateral connectivity in this layer ensures that the action neurons compete with each other in a winner-take-all scheme. Their activity is then used to determine the action (i.e. the direction and velocity) to take at every instant (*Figure 3*).

## Place cells (*Figure 3bii*)

The agent moves in an open field, modelled as a square of side length of 4 a.u. The initial position of the agent in each trial is the centre of the open field, which corresponds to the origin of the cartesian plane. The position of the agent at time $t$ is described by the bi-dimensional vector of its Cartesian coordinates, $\mathbf{x}(t)$. The $11 \times 11 = 121$ place cells are distributed on a grid, at a horizontal and vertical distance of $\sigma = 0.4$ from one another. The spiking activity of place cell $i$ is modeled as an inhomogeneous Poisson process, with parameter $\lambda_i^{pc}(\mathbf{x}(t))$ defined as follows:

$$\lambda_i^{pc}(\mathbf{x}(t)) = \bar{\lambda}^{pc} \exp\left( -\frac{||\mathbf{x}(t) - \mathbf{x}_i||^2}{\sigma^2} \right).$$

The firing rate $\lambda_i^{pc}$ is a function of the distance of the agent from the place cell centre $\mathbf{x}_i$. It is at its maximum, $\bar{\lambda}^{pc} = 400$ Hz, when the agent is located exactly in $\mathbf{x}_i$ and it decreases as the agent moves away. This mechanism simulates a place field, which allows for an accurate representation of the position of the agent in the environment.

## Action neurons (*Figure 3bii*)

Place cells constitute the input to the network, and they project to all action neurons with weights $w^{feed}$. These feed-forward weights are initialized to $w_{in} = 2$ and bounded between $w_{min} = 1$ and $w_{max} = 3$. Action neurons are also connected with each other through synaptic weights $w^{lat}$. The neurons are modeled as zero-order Spike Response Model ($\mathrm{SRM}_0$; *Gerstner, 1995*). The membrane potential of neuron $j$ is given by:

$$u_j(t) = \sum_i \sum_{\bar{t}_i \in F_i^{pc}, t > \hat{t}_j} w_{ji}^{feed} \cdot \epsilon(t - \bar{t}_i) + \sum_{k, k \neq j} \sum_{\bar{t}_k \in F_k^a, t > \hat{t}_j} w_{jk}^{lat} \cdot \epsilon(t - \bar{t}_k) + \chi \Theta(t - \hat{t}_j) \exp\left( -\frac{t - \hat{t}_j}{\tau_m} \right),$$

where $\chi = -5$ mV scales the refractory period; $\hat{t}_j$ is the last postsynaptic spiking time; and $\epsilon$ is the EPSP described by the kernel $\epsilon(t) = \frac{\epsilon_0}{\tau_m - \tau_s} \left( e^{\frac{-t}{\tau_m}} - e^{\frac{-t}{\tau_s}} \right) \Theta(t)$, with $\Theta(t)$ being the Heaviside step function, $\tau_m = 20$ ms, $\tau_s = 5$ ms, $\epsilon_0 = 20$. $F_i^{pc}$ and $F_k^a$ are sets containing, respectively, $\bar{t}_i$ and $\bar{t}_k$, the arrival times of all spikes fired by place cell $i$ and action neuron $k$. Spiking behavior is stochastic and follows an inhomogeneous Poisson process with parameter $\lambda_j(u_j(t))$, which depends on the membrane potential at time $t$. In particular,

$$\lambda_j(u_j(t)) = \lambda_0 \exp\left( \frac{u_j(t) - \theta}{\Delta u} \right),$$

where $\lambda_0 = 60$ Hz, $\Delta u = 2$ mV, $\theta = 16$ mV.

Action neurons represent different directions in the Cartesian plane. Specifically, each action neuron $j$ represents direction $\mathbf{a}_j$, where $\mathbf{a}_j = a_0(\sin(\theta_j), \cos(\theta_j))$, with $\theta_j = \frac{2j\pi}{N}$, $N = 40$ and $a_0 = 0.08$. The lateral connectivity between action neuron $k$ and action neuron $j$ is defined as follows:

$$w_{jk}^{lat} = \frac{w_-}{N} + w_+ \frac{f(j,k)}{N},$$

where $w_- = -300$, $w_+ = 100$ and $f$ is a lateral connectivity function, which is symmetric, positive and increases monotonically with the similarity of the actions. In particular, $f(j,k) = (1 - \delta_{jk}) e^{\zeta \cos(\theta_j - \theta_k)}$, with $\zeta = 20$. Neurons therefore excite each other when they have a similar tuning, and depress otherwise. This ensures that only a few similarly tuned action neurons are active at any given time, making the trajectory of the agent smooth and consistent.

## Action selection (*Figure 3biii*)

The action selection process determines $\mathbf{a}(t)$, the action to take at time $t$, based on the firing rates of the action neurons. The activity of action neuron $j$ is approximated by filtering spike train $Y_j$ with kernel $\gamma$:

$$\rho_j(t) = (Y_j \circ \gamma)(t),$$

where $Y_j = \sum_{\bar{t}_j \in F_j^a} \delta(t - \bar{t}_j)$ and $\gamma(t) = \frac{e^{\frac{-t}{\tau_\gamma}} - e^{\frac{-t}{\nu_\gamma}}}{\tau_\gamma - \nu_\gamma} \Theta(t)$, with $\tau_\gamma = 50$ ms and $\nu_\gamma = 20$ ms.

If each action neuron $j$ represents direction $\mathbf{a}_j$ and has an estimated firing rate $\rho_j(t)$, then the action $\mathbf{a}(t)$ is the average of all the directions encoded, weighted by their respective firing rates:

$$\mathbf{a}(t) = \frac{1}{N} \sum_j \rho_j(t) \mathbf{a}_j,$$

where $N = 40$ is the total number of action neurons. This decision making mechanism allows the agent to move in any direction, making the action space effectively continuous.

## Navigation details (*Figure 3biv*)

Once action $\mathbf{a}(t)$ has been determined, the update for the position of the agent is:

$$\Delta \boldsymbol{x}(t) = \begin{cases} \mathbf{a}(t) & \text{if } x(t+1) \text{ in the square} \\ d \cdot \boldsymbol{u}(\boldsymbol{x}(t)) & \text{otherwise} \end{cases}$$

The agent therefore normally moves with instantaneous velocity $\mathbf{a}(t)$. When the agent tries to surpass the limits of the field, it is instantly bounced back by a distance $d = 0.01$. The unit vector $\mathbf{u}(\mathbf{x}(t))$ points in the direction opposite to the boundary. To avoid large boundary effects, the feed-forward weights between place cells on the boundaries and action neurons that code for a direction $\mathbf{a}_j$ outside of the field are set to zero.

The agent is free to explore the environment for a maximum duration of $t_{max} = 15$ s. If it finds the reward at a time $t_{rew} < T_{max}$, the trial is terminated earlier, precisely at time $t = T_{rew} + 300$ ms. The extra time mimics consummatory behavior, navigation is thus paused during this interval (i.e. place cells activity is set to zero). During the first 20 trials, the reward can be found in the circular goal area centered in $c_1 = (1.5, 1.5)$ with radius $r_1 = 0.3$. In trials 21 to 40, the goal area moves to center $c_2 = (-1.5, -1.5)$, but maintains the same shape and size. The effect of the inter-trial interval is modeled by resetting all activity.

## Synaptic plasticity and learning

The synaptic weights between place cells and action neurons play a fundamental role in defining a policy for the agent. Plasticity is essential for the agent to learn to navigate the open field and is implemented in a way that follows the experimental results presented in *Figures 1* and *2* and *Brzosko et al., 2015*. The synaptic changes combine the modified STDP rule (*Figure 3bi*) and an eligibility trace that allows for delayed updates.

In particular, the total weight update is:

$$\Delta w_{ji}(t) = \eta A \left( \left( \sum_{\bar{t}_i \in F_i^{pc}} \sum_{\bar{t}_j \in F_j^a} W(\bar{t}_j - \bar{t}_i) \right) \circ \psi \right)(t),$$

where $\eta$ is the learning rate, $A$ emulates the effect of the different neuromodulators, $W$ is the STDP window and $\psi$ is the eligibility trace. $F_i^{pc}$ and $F_j^a$ are sets containing $\bar{t}_i$ and $\bar{t}_j$, respectively, the arrival times of all spikes fired by place cell $i$ and action neuron $j$.

The basic STDP window is $W(x) = e^{-\frac{|x|}{\tau}}$, with $\tau = 10$ ms. This function is always symmetric and positive, but the sign of the final weight change is determined by the neuromodulators at the synapse:

$$A = \begin{cases} A_{ACh} = -1 & -\text{DA}, +\text{ACh} \\ 0 & -\text{DA}, -\text{ACh} \\ A_{DA} = +1 & +\text{DA}, \pm\text{ACh} \end{cases}$$

Dopamine is assumed to be released simultaneously at all synapses whenever a reward is reached. All weight changes are gated by neuromodulators ($A = 0$ when all neuromodulators are absent). The learning rate $\eta$ also depends on neuromodulators:

$$\eta = \begin{cases} 0.002 & +\text{ACh}, -\text{DA} \\ 0 & -\text{ACh}, -\text{DA} \\ 0.01 & \pm\text{ACh}, +\text{DA} \end{cases}$$

The weight change due to STDP is convoluted with an eligibility trace $\psi$, modeled as an exponential decay $\psi(t) = e^{-\alpha \frac{t}{\tau_e}} \Theta(t)$, with $\tau_e = 2$ s and $\alpha = \begin{cases} 1 & +\text{DA} \\ 0 & -\text{DA} \end{cases}$. The eligibility trace keeps track of the active synapses and allows for a delayed update of the synaptic strength. Variable $\alpha$ in the exponent acts as a flag and ensures that the eligibility trace is active with dopamine only ($\alpha = 1$).

Two sets of simulations (1000 simulations each) were performed. In the first set (control: $-$ ACh), no interaction with acetylcholine was assumed. The weights were therefore potentiated only when the agent found the reward ($A = 1$, $\alpha = 1$) and left unchanged otherwise ($A = 0$). In the second set of simulations (test: $+$ ACh), acetylcholine was present throughout the task. The weights were updated online ($A = -1$, $\alpha = 0$). When no reward was found before the end of the trial, weights were depressed. They were otherwise potentiated when reward was found ($A = 1$, $\alpha = 1$).

## Predictions

We tested the effect of varying the amplitudes of the STDP learning windows under dopaminergic and cholinergic modulation, $A_{DA}$ and $A_{ACh}$, on the agent's behavior. We ran simulations under Small, Medium and Large magnitudes of dopamine and acetylcholine effects, resulting in five different conditions (1000 simulations each). The exact parameters used can be found in the table below.

| | Magnitude of neuromodulation | $A_{ACh}$ | $A_{DA}$ | Figure |
|---|---|---|---|---|
| Dopamine | Small | −1 | 0.1 | *Figure 4avi*; *Figure 4—figure supplement 2* |
| | Medium | −1 | 1 | |
| | Large | −1 | 3 | |
| Acetylcholine | Small | −0.5 | 1 | *Figure 4bvi*; *Figure 4—figure supplement 3* |
| | Medium | −1 | 1 | |
| | Large | −1.5 | 1 | |

## Dopamine-modulated standard asymmetric STDP curve

We also compared our symmetric learning windows with standard asymmetric STDP curves (*Figure 4—figure supplement 4*). The total weight update with this rule is

$$\Delta w_{ji}(t) = \eta \left( \left( \sum_{\bar{t}_i \in F_i^{pc}} \sum_{\bar{t}_j \in F_j^a} W_2\left(\bar{t}_j - \bar{t}_i\right) \right) \circ \psi \right)(t),$$

where $\eta = 0.01$ is the learning rate, $W_2$ is the STDP window and $\psi$ is the eligibility trace (as defined above). $F_i^{pc}$ and $F_j^a$. are sets containing $\bar{t}_i$ and $\bar{t}_j$, respectively, the arrival times of all spikes fired by place cell $i$ and action neuron $j$.

The spike timing plasticity rule was implemented as follows

$$W_2(x) = \begin{cases} A_{pre-post} e^{-\frac{x}{\tau}} & \text{if } x > 0 \\ \frac{1}{2}\left(A_{pre-post} - A_{post-pre}\right) & \text{if } x = 0 \\ A_{post-pre} e^{\frac{x}{\tau}} & \text{if } x < 0 \end{cases}$$

The integral of the learning window determines if the agent learns, unlearns or does not learn (*Figure 4—figure supplement 4*). We therefore considered three different parameter sets: (i) positive integral ($A_{pre-post} = 2$, $A_{post-pre} = -1$); (ii) zero integral ($A_{pre-post} = 1$, $A_{post-pre} = -1$); (iii) negative integral ($A_{pre-post} = 1$, $A_{post-pre} = -2$). The time constant was identical for the two sides of the window and was taken to be $\tau = 10$ ms. We ran 1000 simulations for each parameter set.

## Acknowledgements

This research was supported by an MRC studentship to ZB, an EPSRC studentship to SZ, BBSRC grants to CC and OP (BB/N013956/1 and BB/N019008/1), and Wellcome Trust Awards to WS and CC (095495 and 200790/Z/16/Z).

## Additional information

### Competing interests

WS: Reviewing Editor, eLife. The other authors declare that no competing interests exist.

### Funding

| Funder | Grant reference number | Author |
| --- | --- | --- |
| Medical Research Council | Studentship | Zuzanna Brzosko<br>Wolfram Schultz<br>Ole Paulsen |
| Engineering and Physical Sciences Research Council | Studentship | Sara Zannone<br>Claudia Clopath |
| Wellcome | 095495 | Wolfram Schultz |
| Biotechnology and Biological Sciences Research Council | BB/N013956/1 | Claudia Clopath |
| Wellcome | 200790/Z/16/Z | Claudia Clopath |
| Biotechnology and Biological Sciences Research Council | BB/N019008/1 | Ole Paulsen |

The funders had no role in study design, data collection and interpretation, or the decision to submit the work for publication.

### Author contributions

ZB, Conceptualization, Data curation, Formal analysis, Investigation, Visualization, Methodology, Writing—original draft, Writing—review and editing; SZ, Software, Investigation, Methodology, Writing—review and editing; WS, Conceptualization, Supervision, Funding acquisition, Writing—review and editing; CC, Conceptualization, Resources, Software, Supervision, Funding acquisition, Methodology, Project administration, Writing—review and editing; OP, Conceptualization, Resources, Supervision, Funding acquisition, Methodology, Writing—original draft, Project administration, Writing—review and editing

### Author ORCIDs

Zuzanna Brzosko, http://orcid.org/0000-0003-0654-2655
Sara Zannone, http://orcid.org/0000-0002-7189-2435
Wolfram Schultz, http://orcid.org/0000-0002-8530-4518
Claudia Clopath, http://orcid.org/0000-0003-4507-8648
Ole Paulsen, http://orcid.org/0000-0002-2258-5455

### Ethics

Animal experimentation: The research was performed under the Animals (Scientific Procedures) Act 1986 Amendment Regulations 2012 following ethical review by the University of Cambridge Animal Welfare and Ethical Review Body (AWERB). The animal procedures were authorised under Project licence PPL 70/8892.

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
