## [Decision Letter]

Thank you for submitting your article "Sequential neuromodulation of Hebbian plasticity offers mechanism for effective reward-based navigation" for consideration by *eLife*. Your article has been reviewed by two peer reviewers, and the evaluation has been overseen by a Reviewing Editor and Eve Marder as the Senior Editor. The following individuals involved in review of your submission have agreed to reveal their identity: Hitoshi Morikawa (Reviewer #1); Guoqiang Bi (Reviewer #2).

The reviewers have discussed the reviews with one another and the Reviewing Editor has drafted this decision to help you prepare a revised submission.

This work is a follow-up of the authors' previous work on the modulation of spike time dependent plasticity by dopamine. The authors found that the neuromodulator, acetylcholine, facilitates synaptic depression when applied prior to pairing stimulation apparently through muscarinic receptor activation. This depression can be reversed to long-term potentiation by subsequent application of dopamine, very likely via cAMP signaling. Implementing this bi-directionally modulated spike time dependent plasticity rule in a computational model yielded flexible learning of navigation towards changing target locations.

Both reviewers judged the study as interesting, linking synaptic plasticity rules with behavioral learning. Following the reviewers' comments the BRE would like to ask you to provide evidence that the timing of ACh and DA application is important for the observed changes in plasticity (ACh needs to be present during pairing and DA needs to be present after pairing; see reviewer #1). Reviewer #1 asked for additional figures and data analysis, which are described below in a point-by-point manner. The second reviewer wishes that you compare the effectiveness of the new learning rule with the classic STDP-based learning rule from e.g. Gerstner and Abbott 1997. This reviewer also asked for some direct predictions, e.g. the consequences of manipulating ACh receptors, which can be tested experimentally. We are including those reviews in their entirety in case this is helpful to you.

*Reviewer #1:*

1) The entire timing of STDP in the presence of ACh is not characterized. For example, plasticity induction in the presence of ACh could lead to LTD regardless of the pre-post timing.

2) The significance of the timing of ACh and DA application should be examined. It is assumed that ACh needs to be present during pairing and DA needs to be present after pairing but this necessity has not been explicitly examined (e.g., ACh present only after pairing).

3) STDP timing curves under basal condition, in DA alone, in ACh alone, and in DA + ACh in the model should be illustrated as a figure. These timing curves should be discussed with respect to those demonstrated in the field (e.g., see Figure 2 in Feldman DE, Neuron 75, 556-571, 2012).

4) Eligibility traces with decay kinetics should also be illustrated as a figure and discussed in comparison to DA action on STDP in brain slice. This will clearly illustrate the two timing rules, msec order timing for STDP and slower timing for DA action.

*Reviewer #2:*

The experiments were well designed and carried out. My main question is about their computational implications. The effects of ACh and dopamine on STDP are certainly interesting. However, they also "disable" the temporal asymmetry of STDP (as suggested in Figure 3), a feature that was used in earlier modeling studies to accomplish learning of temporal sequences and navigational tasks.

1) Can the authors compare the effectiveness of this new learning rule with classic STDP-based learning (e.g. Gerstner and Abbott 1997 as subsequent models)?

2) Furthermore, can the authors offer some direct predictions, e.g. the consequences of manipulating ACh receptors, which can be tested by experiments?

---

## [Author Response]

*[…] Both reviewers judged the study as interesting, linking synaptic plasticity rules with behavioral learning. Following the reviewers' comments the BRE would like to ask you to provide evidence that the timing of ACh and DA application is important for the observed changes in plasticity (ACh needs to be present during pairing and DA needs to be present after pairing; see reviewer #1). Reviewer #1 asked for additional figures and data analysis, which are described below in a point-by-point manner. The second reviewer wishes that you compare the effectiveness of the new learning rule with the classic STDP-based learning rule from e.g. Gerstner and Abbott 1997. This reviewer also asked for some direct predictions, e.g. the consequences of manipulating ACh receptors, which can be tested experimentally. We are including those reviews in their entirety in case this is helpful to you.*

*Reviewer #1:*

*1) The entire timing of STDP in the presence of ACh is not characterized. For example, plasticity induction in the presence of ACh could lead to LTD regardless of the pre-post timing.*

To address this comment we carried out an additional set of experiments, in which exogenous acetylcholine (1 µM) was bath-applied for 10 minutes from 1-2 minutes before and throughout a pairing protocol with extended spike timing intervals of either Δt = +50 ms or Δt = −50 ms. Application of acetylcholine during these pairing protocols did not induce a significant change in synaptic weights (Δt = +50 ms: 84 ± 11%, *t*(6) = 1.5, *P* =0.1852 vs. 100%, *n* = 7; Δt = −50 ms: 113 ± 15%, *t*(4) = 0.8, *P* =0.4409 vs. 100%, *n* = 6). This has now been presented in Figure 1 and emphasized in the first paragraph of the Results and Discussion. Of note, application of acetylcholine also had no significant effect on the Schaffer collateral basal transmission in the absence of pairing with postsynaptic action potentials (Figure 1).

By including these two additional spike timing intervals in Figure 1, we have now characterized an expanded time window for presynaptic-postsynaptic spike pair interactions at the Schaffer collateral pathway in the presence of acetylcholine. Together, our data implies that activation of cholinergic receptors during the coordinated spiking activity with narrow time intervals between presynaptic and postsynaptic spikes (−50 ms < Δt < +50 ms) biases STDP towards synaptic depression.

*2) The significance of the timing of ACh and DA application should be examined. It is assumed that ACh needs to be present during pairing and DA needs to be present after pairing but this necessity has not been explicitly examined (e.g., ACh present only after pairing).*

To address this comment we examined whether acetylcholine applied after the pairing protocol influences the expression of STDP. In contrast to dopamine, which can retroactively convert negative spike timing t-LTD into t-LTP (Brzosko et al., 2015) and acetylcholine-facilitated positive spike timing LTD into t-LTP (current study), acetylcholine applied after a pairing protocol with Δt = +10 ms did not affect the development of t-LTP (153 ± 15%, *t*(3) = 3.5, *P* =0.0249 vs. 100%, *n* = 5). This result has now been presented in Figure 2—figure supplement 1 and emphasized in the third paragraph of the Results and Discussion).

Notably, this finding highlights the importance of timing and context of neuromodulation, and we have illustrated the possible behavioral relevance of these effects with an updated schematic in Figure 3 (see Comment 4).

*3) STDP timing curves under basal condition, in DA alone, in ACh alone, and in DA + ACh in the model should be illustrated as a figure. These timing curves should be discussed with respect to those demonstrated in the field (e.g., see Figure 2 in Feldman DE, Neuron 75, 556-571, 2012).*

We have performed experiments to investigate STDP curves under (i) basal condition and application of: (ii) acetylcholine alone; (iii) dopamine alone; (iv) acetylcholine and dopamine during the positive and negative pairing protocols.

i) The STDP curve under basal conditions at the Schaffer collateral pathway is presented in Figure 1 (and inBrzosko et al., 2015). In accordance with previous studies in the hippocampus (Bi & Poo, 1998; Zhang et al., 2009), the control STDP curve under our experimental conditions shows a classical Hebbian, asymmetric time window.

ii) Application of acetylcholine during the coordinated spiking activity biases the STDP curve towards depression (Figure 1).

iii) Application of dopamine during the coordinated spiking activity biases the STDP curve towards potentiation. This is now presented in Figure 1—figure supplement 1 and summarised in the first paragraph of the Results and Discussion.

iv) Co-application of acetylcholine and dopamine during the pairing protocol with Δt = −20 ms results in synaptic depression. Activation of cholinergic and dopaminergic receptors during pairing protocol with Δt = +10 ms leads to an initial synaptic depression followed by a gradual reversal of the synaptic depression back towards baseline. This data is now presented in Figure 1—figure supplement 1 and summarised in the first paragraph of the Results and Discussion. Crucially, the STDP curve under co-application of acetylcholine and dopamine is different to our novel learning rule under sequential modulation by acetylcholine and dopamine, which is the basis of our computational model.

In the computational simulations, we have now compared the two models using sequentially modulated symmetric STDP rules, with and without cholinergic depression, now presented in Figure 3BI,as well as the standard reward-modulated STDP model (Figure 4—figure supplement 4).

*4) Eligibility traces with decay kinetics should also be illustrated as a figure and discussed in comparison to DA action on STDP in brain slice. This will clearly illustrate the two timing rules, msec order timing for STDP and slower timing for DA action.*

We would like to thank for the suggestion of incorporating such a schematic, which is now presented inFigure 3.

*Reviewer #2:*

*The experiments were well designed and carried out. My main question is about their computational implications. The effects of ACh and dopamine on STDP are certainly interesting. However, they also "disable" the temporal asymmetry of STDP (as suggested in Figure 3), a feature that was used in earlier modeling studies to accomplish learning of temporal sequences and navigational tasks.*

*1) Can the authors compare the effectiveness of this new learning rule with classic STDP-based learning (e.g. Gerstner and Abbott 1997 as subsequent models)?*

To address this question, we changed our learning rule to a classic asymmetrical STDP curve (i.e., with positive pre-post side, and negative post-pre side), while keeping the experimental protocol and the other parameters unvaried. The results reveal that, at least in our model (Frémaux et al., 2013), the exact spike timing and therefore the asymmetry of the STDP learning window are rather unimportant. It is the integral that determines the agents’ performance (Figure 4—figure supplement 4; Results and Discussion, sixth paragraph). In addition, while classic reward modulated STDP only allows the agent to either learn or unlearn, our sequentially neuromodulated learning rule can switch flexibly between the two and thus performs better in changing environments (Results and Discussion, sixth paragraph).

*2) Furthermore, can the authors offer some direct predictions, e.g. the consequences of manipulating ACh receptors, which can be tested by experiments?*

We performed simulations with different magnitudes of dopaminergic potentiation and cholinergic depression and tested their impact on the agents’ performance. Varying the magnitude of dopamine effect at the synapses had a larger effect on learning, but only a small difference was observed between Medium and Large conditions, likely due to a saturation effect (Figure 4AVI and Figure 4—figure supplement 2). Acetylcholine mainly modulated unlearning, with larger magnitudes of acetylcholine effect yielding faster unlearning (Figure 4BVI and Figure 4—figure supplement 3). These results are also described in the fifth paragraph of the Results and Discussion.